# High-frequency terahertz stimulation alleviates neuropathic pain by inhibiting the pyramidal neuron activity in the anterior cingulate cortex of mice

Wenyu Peng[1†], Pan Wang[2†], Chaoyang Tan[2†], Han Zhao[2], Kun Chen[2], Huaxing Si[2], Yuchen Tian[1], Anxin Lou[2], Zhi Zhu[3], Yifang Yuan[4], Kaijie Wu[5,6]*, Chao Chang[4,7]*, Yuanming Wu[1]*, Tao Chen[2]*

[1]Department of Biochemistry and Molecular Biology, School of Basic Medicine, The Fourth Military Medical University, Xi'an, China; [2]Department of Anatomy, Histology and Embryology and K.K. Leung Brain Research Centre, The Fourth Military Medical University, Xi'an, China; [3]Laboratory of Optical Technology and Instrument for Medicine, Ministry of Education, College of Optical-Electrical and Computer Engineering, University of Shanghai for Science and Technology, Shanghai, China; [4]Innovation Laboratory of Terahertz Biophysics, National Innovation Institute of Defense Technology, Beijing, China; [5]Information Materials and Intelligent Sensing Laboratory of Anhui Province, Anhui University, Hefei, China; [6]School of Electronic and Information Engineering, Anhui University, Hefei, China; [7]School of Physics, Peking University, Beijing, China

*For correspondence:
23109@ahu.edu.cn (KW);
gwyzlzssb@pku.edu.cn (CC);
wuym@fmmu.edu.cn (YW);
taochen1@foxmail.com (TC)

[†]These authors contributed equally to this work

**Abstract** Neuropathic pain (NP) is caused by a lesion or disease of the somatosensory system and is characterized by abnormal hypersensitivity to stimuli and nociceptive responses to non-noxious stimuli, affecting approximately 7–10% of the general population. However, current first-line drugs like non-steroidal anti-inflammatory agents and opioids have limitations, including dose-limiting side effects, dependence, and tolerability issues. Therefore, developing new interventions for the management of NP is urgent. In this study, we discovered that the high-frequency terahertz stimulation (HFTS) at approximately 36 THz effectively alleviates NP symptoms in mice with spared nerve injury. Computational simulation suggests that the frequency resonates with the carbonyl group in the filter region of Kv1.2 channels, facilitating the translocation of potassium ions. In vivo and in vitro results demonstrate that HFTS reduces the excitability of pyramidal neurons in the anterior cingulate cortex likely through enhancing the voltage-gated K$^+$ and also the leak K$^+$ conductance. This research presents a novel optical intervention strategy with terahertz waves for the treatment of NP and holds promising applications in other nervous system diseases.

## eLife assessment

Peng et al. reported **important** findings that 36THz high-frequency terahertz stimulation (HFTS) could suppress the activity of pyramidal neurons by enhancing the conductance of voltage-gated potassium channels. The significance of the findings in this paper is that chronic pain remains a significant medical problem, and there is a need to find non-pharmacological interventions for treatment. The authors present **convincing** evidence that high-frequency stimulation of the anterior cingulate cortex can alter neuronal activity and improve sensory pain behaviors in mice.

**eLife digest** Up to 1 in 10 people are estimated to experience neuropathic pain, a particularly challenging form of chronic pain where nerve damage causes extreme sensitivity to everyday stimuli. Current treatments often rely on painkiller drugs that can lead to serious side effects as well as dependency issues. New and effective interventions are therefore necessary.

One radically different approach is the use of 'terahertz' waves, a type of electromagnetic radiation that has the ability to affect the chemical bonds holding molecules together. In fact, previous research has shown that specific frequencies of terahertz waves can modify the activity of certain proteins. With this technique, it may therefore be possible to disrupt voltage-dependent potassium channels, a type of proteins which help to regulate nerve cell activity and is a possible target for pain therapy.

To explore this approach, Peng, Wang, Tan et al. investigated whether high-frequency terahertz stimulation that targets potassium ion channels could reduce neuropathic pain in mice. The animals, which had undergone surgery recreating nerve damage, were implanted with a device that allowed the delivery of terahertz waves into a brain region vital for regulating pain sensations. Experiments showed that delivering 36 terahertz radiations changed important ion channel properties (such as how easily they would allow ions to pass through), decreasing neuron activity and raising the pain threshold of the mice.

This finding indicates that, with further development, terahertz frequency stimulation could become a new, non-drug method to manage neuropathic pain. Additional research will be needed to see if terahertz waves could also be applied to other neurological disorders influenced by ion channel activity.

## Introduction

NP refers to a debilitating chronic pain condition, which is often a consequence of nerve injury or of the diseases such as cancer, diabetes mellitus, infection, autoimmune disease, and trauma (*Baron et al., 2010*; *Shan et al., 2023*). The symptoms of NP include spontaneous pain, hyperalgesia, and mechanical allodynia. Unfortunately, NP is often resistant to currently available drug treatments, including non-steroidal anti-inflammatory drugs and even opioids (*Jensen and Finnerup, 2014*). More evidences reveal that NP is not merely a symptom of a disease but rather an expression of pathological operations of the nervous system (*Costigan et al., 2009*). Therefore, developing new therapeutic technology aimed at these underlying mechanisms for pain relief represents a considerable challenge.

Compared with the limitations of chemical-based drug research, physics-based treatment offers a new concept and opportunity for intervening in NP. Optogenetics, as an interdisciplinary approach, has demonstrated therapeutic potential in NP. However, the limitations of viral vector delivery systems in humans are well-known (*Liu and Wang, 2019*). Recently, evidence has emerged suggesting that high-frequency terahertz (THz) photons directly resonate with molecules, thereby regulating corresponding biological functions. For instance, our previous study demonstrated that a 34.88 THz wave resonates with Aβ protein, disrupting the process of fibril formation (*Peng et al., 2023*). Li et al. discover that the band of 42.55 THz resonates with the stretching mode of either the –COO- or the –C=O group significantly enhancing the $Ca^{2+}$ conductance (*Li et al., 2021*). Zhu et al. conclude that 48.2 THz photons greatly increase the permeability of the sodium channel by a factor of 33.6 through breaking the hydrated hydrogen bonding network between the hydrosphere layer of the ions and the carboxylate groups (*Zhao et al., 2023*). Additionally, the frequency of 53.5 THz has been reported to enhance the voltage-gated $K^+$ currents, which modulate the startle response and associative learning (*Zhang et al., 2021a*; *Liu et al., 2021*). These studies strongly prompt us to the potential application of THz photons in the treatment of neuropathic pain by targeting the ion channels (*Trimmer, 2014*).

The anterior cingulate cortex plays a crucial role in pain regulation (*Bliss et al., 2016*). Our previous research has demonstrated that nociceptive information resulting from nerve injury is transmitted to the ACC (*Tsuda et al., 2017*). This region exhibits pre- and postsynaptic long-term plasticity (LTP), which contributes to chronic pain and associated negative emotions (*Chen et al., 2014*; *Koga et al., 2015*). Furthermore, descending projection pathways from the ACC enhance the neuronal activity of the spinal dorsal horn (SDH) and regulate nociceptive sensory transmission (*Chen et al., 2018b*). Brain imaging and MRI studies also provide evidence of hyperexcitability in the ACC during both acute and

chronic pain (*Alles and Smith, 2018*; *Bliss et al., 2016*). Specifically, the activity of pyramidal cells in the ACC is directly correlated with the expression of chronic pain (*Zhu et al., 2022*; *Li et al., 2010*). Optogenetic excitation of ACC pyramidal cells induces pain, while their inhibition leads to analgesia (*Kang et al., 2015*). Therefore, targeting the cortical regions of the ACC and inhibiting the activity of ACC pyramidal neurons may hold promise as a strategy for treating NP (*Fuchs et al., 2014*; *Lançon et al., 2021*; *Um et al., 2019*; *Tan et al., 2015*).

Neuronal excitability is influenced by various types of voltage-gated ion channels and among them, voltage-dependent potassium (Kv) channels, as one of the important physiological regulators of neuronal membrane potentials, has been proposed as potential target candidates for pain therapy (*Takeda et al., 2011*; *Du and Gamper, 2013*; *Sakai et al., 2017*). Zhao et al. have reported that enhancing Kv currents in injured dorsal root ganglion (DRG) neurons alleviates neuropathic pain (*Zhao et al., 2017*). Additionally, Fan et al. have demonstrated that lumbar $(L)_5$ spinal nerve ligation (SNL) leads to a time-dependent decrease in Kv1.2-positive neurons in the ipsilateral $L_5$ DRG. However, rescuing Kv1.2 expression in the injured L5 DRG attenuates the development and persistence of pain hypersensitivity (*Fan et al., 2014*). These findings highlight the potential of targeting Kv channels as a therapeutic approach for managing pain.

In this study, we investigated the effects of HFTS on the Kv model. By analyzing the absorbance spectra of Kv1.2 channels, we observed a significant response to photons with a frequency of approximately 36 THz. This frequency modulates the resonance of the carbonyl group in the Kv1.2 structure, affecting the action potential waveform and frequency as demonstrated through simulations. Subsequently, we conducted in vivo multi-channel recordings and in vitro patch recordings to confirm the activation effect of HFTS on $K^+$ conductance and its inhibition of neuronal activity in the ACC pyramidal cells. Importantly, the application of HFTS resulted in a significant reduction in pain behavior in mice with spared nerve injury (SNI).

## Results
### HFTS attenuates the generation of action potential through molecular dynamics simulation

To identify a specific terahertz (THz) frequency capable of modulating a major subset of voltage-gated potassium (Kv) channels, we developed an integrated model comprising both mouse Kv channels (Protein Data Bank [PDB] ID: 3LUT) and $Na^+$ channels (PDB ID: 3RVY) (*Figure 1a* and Supplement Fig. S1). We conducted a comprehensive analysis of the spectral absorption characteristics within the THz frequency range. Our results revealed a pronounced absorption peak at approximately 36 THz for the potassium channel, which exhibited a considerable board band compared to the absence of a corresponding peak for the sodium channel (*Figure 1a*). This indicates a preferential and resonant absorption of photons at the ~36 THz frequency by the potassium channel. We then tested the possible kinetic changes of Kv1.2, the typical and widely distributed Kv channel in the central nervous system (*Tsantoulas and McMahon, 2014*), following the absorption of these THz photons. Our findings indicated a significant kinetic change in the -C=O groups of the channel filter structure, as evidenced by an expansion of its van der Waals radius by approximately 0.5 Å (*Figure 1b*). Furthermore, during exposure to THz photons, the conductance of the potassium ion channel exhibited an almost linear increase with the intensity of the THz field, while the conductance of sodium ions remained largely unchanged (*Figure 1c*). Interestingly, under terahertz photonic influence, the cortical neurons model showed significantly decreased in discharge (*Figure 1d*). We performed a detailed waveform analysis of the action potentials, including parameters such as full width at half maximum (FWHM) and firing frequency (*Figure 1e*). Our observations revealed that the FWHM of action potentials in the THz-exposed group decreased to 95% of the control group (*Figure 1f*, red column), and the firing frequency experienced a reduction of approximately 70% after THz photon stimulation (*Figure 1f*, green column). These results collectively suggest that THz photons primarily attenuate neuronal firing activity by increasing potassium ion conductance, thereby modulating neuronal excitability.

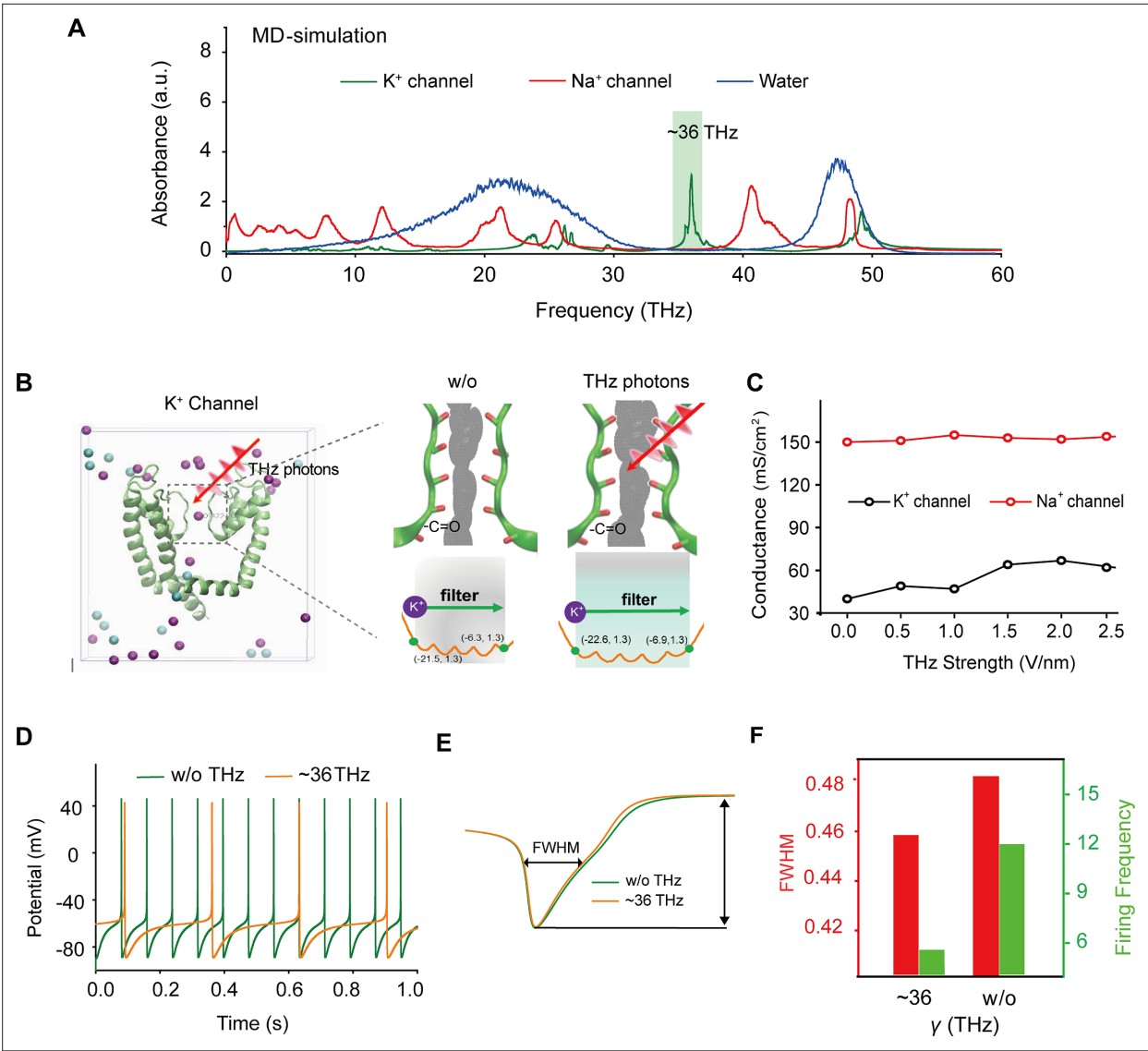

**Figure 1.** Specific frequency terahertz (THz) photons resonate the voltage-dependent potassium (Kv) channel and decrease the AP firing rate in cortical neurons through molecular dynamics simulation. (**a**) Absorbance spectra of voltage-gated potassium/sodium ion channels and the bulk water. (**b**) The dynamic attributes of the Kv1.2 filter structure in pre- and post-exposure to HFTS. Purple balls represent the $K^+$, and blue balls represent the $Cl^-$. (**c**) The alterations in potassium/sodium ion conductance are consequent to the influence of HFTS. (**d**) Changes in the firing rate of APs of cortical neuron models before and after HFTS. (**e**) The FWHM of an AP pre- and post-HFTS. (**f**) Changes in FWHM and firing frequency with or without HFTS. HFTS, high-frequency terahertz stimulation. AP, action potential. FWHM, Full Wide of Half Maximum.

The online version of this article includes the following source data and figure supplement(s) for figure 1:

**Source data 1.** Shows the videos of Kv channel transporting K+ with and without HFTS.

**Figure supplement 1.** Sodium (Na+) channels (PDB ID: 3RVY) developed in this study with a clean view.

## HFTS enhances voltage-gated $K^+$ currents and leaks $K^+$ currents of pyramidal neurons in the ACC

To investigate the impact of HFTS on voltage-gated potassium/sodium (Kv/Nav) channels, which play a crucial role in action potential generation and waveform, we conducted whole-cell patch recording from layer-5 pyramidal neurons (PYR[ACC]) in acute slices of the anterior cingulate cortex (ACC) in mice with spared nerve injury (SNI) (*Figure 2a*). Initially, we examined the Nav current by applying a series of test pulses (from –80 to –10 mV) with a command voltage of –100 mV (20 ms) (*Figure 2b*). Upon illumination with ~36 THz photons (0.3±0.05 mW) for durations of 5, 10, and 20 min, we observed

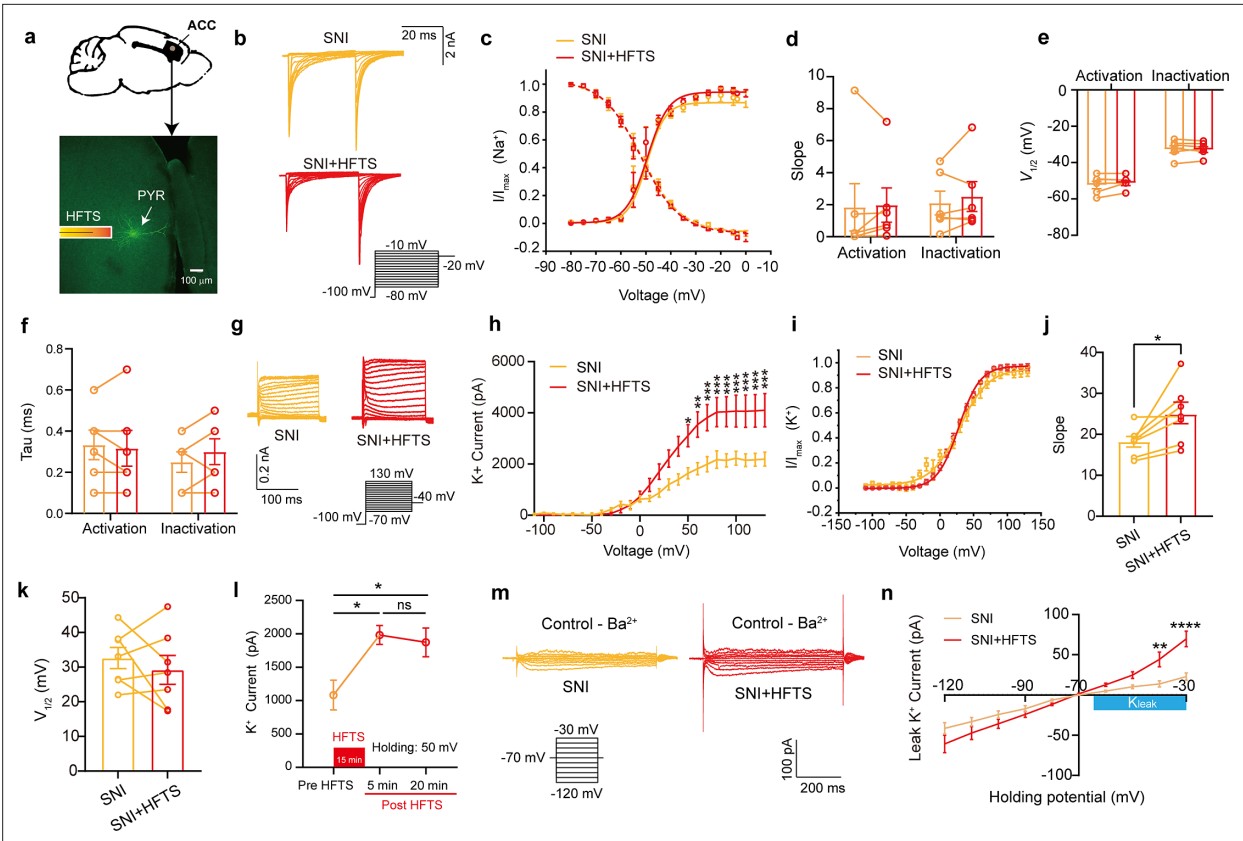

**Figure 2.** High-frequency terahertz stimulation (HFTS) enhances voltage-dependent potassium (Kv) channels and $K_{Leak}$ currents of pyramidal neurons in spared nerve injury (SNI) mouse in vitro. (**a**) Anatomical location of ACC region in mice and a recorded pyramidal (PYR) neuron (biocytin-labeled, green). (**b**) Representative Nav currents without (orange) or with HFTS (red) under the given step voltage protocol. (**c**) The activation and inactivation curves of Nav currents with and without HFTS. (**d-f**) The corresponding slopes of the activation and inactivation curves (**d**), the comparison of the half-activation and inactivation voltages (**e**), and the time constants (tau) of half-activation voltage/half-inactivation voltage (**f**). (**g**) Representative Kv currents are evoked by a series of step voltages (inset) without (orange) or with HFTS (red). (**h**) I-V plots constructed from the values of traces shown in (**g**). (SNI vs. SNI +HFTS: $F_{(1, 10)}$=6.846, p<0.0001, $n_{SNI}$ = 6, $n_{SNI+HFTS}$ = 6; Two-way ANOVA followed by post hoc comparison using the Šídák's multiple comparisons test). (**i**) The activation curves of the Kv currents with and without HFTS. (**j**) The corresponding slopes of the activation curves (SNI vs. SNI +HFTS: t=5.872, p=0.0011, n=7, unpaired t-test. **p<0.01, ****p<0.0001). (**k**) The half-activation voltages of the activation curves. (**l**) Changes in the impact of Kv current post-HFTS. ($F_{(4, 15)}$=4.19, p=0.0178, n=4; One-way ANOVA followed by post hoc comparison using the Šídák's multiple comparisons test). (**m**) Representative $K_{Leak}$ currents evoked by a series of step voltages (inset) without (orange) or with HFTS (red). (**n**) I-V plots constructed from the values of traces shown in (**m**). (SNI vs. SNI +HFTS: $F_{(1, 12)}$=1.688, p=0.2182, $n_{SNI}$ = 7, $n_{SNI+HFTS}$ = 7; Two-way ANOVA followed by post hoc comparison using the Šídák's multiple comparisons test).

The online version of this article includes the following source data for figure 2:

**Source data 1.** Original data for *Figure 2*.

that HFTS had no significant effect on the activation and inactivation curve slope, half-activation and half-inactivation voltage and time constants (tau) for half-activation and half-inactivation voltage (***Figure 2c–f***). These findings indicate that HFTS does not affect Nav channel-mediated currents. Subsequently, we investigated the influence of HFTS on Kv currents by applying a series of test pulses (100 ms) ranging from –70 to +130 mV with a command voltage of –100 mV (***Figure 2g***). Our results demonstrated that the application of HFTS induced a significant increase in the amplitude of $K^+$ currents (***Supplementary file 1a***) and an enhanced slope of the current-voltage characteristic (I-V) curve (***Figure 2h–j***), without affecting the half-activation voltage (***Figure 2k***). Furthermore, to assess the duration of neuronal effects induced by HFTS (15 min), we examined Kv currents at 5 min and 20 min post-HFTS. It was observed that the Kv current were enhanced 20 min post-HFTS (***Figure 2l***, ***Supplementary file 1b***). Simultaneously, we studied the effect of HFTS on $K_{Leak}$ currents by applying a series of test pulses (400ms) ranging from –120 to –30 mV followed by a command voltage of –70 mV (***Figure 2m***). Currents recorded at and above –65 mV membrane potential were analyzed for potential

$K_{Leak}$ currents (*Khandelwal et al., 2021*). We found that HFTS also induced a significant increase of $K_{Leak}$ currents at holding potential at –40 and –30 mV (*Figure 2n*; *Supplementary file 1c*). These experiments demonstrated that HFTS at approximately 36 THz not only influenced Kv currents but also affected $K_{Leak}$ channel activity, resulting in an acceleration of potassium ion flow and an increase in potassium conductance in PYR[ACC] neurons. Importantly, these experimental findings were consistent with the results obtained from molecular dynamics analysis.

## HFTS reduces the spike frequency of pyramidal neurons in the ACC

We proceeded to investigate the impact of the specific resonant frequency of THz photons on the excitability of PYR[ACC] neurons in SNI and sham mice. Using whole-cell current-clamp recording, we

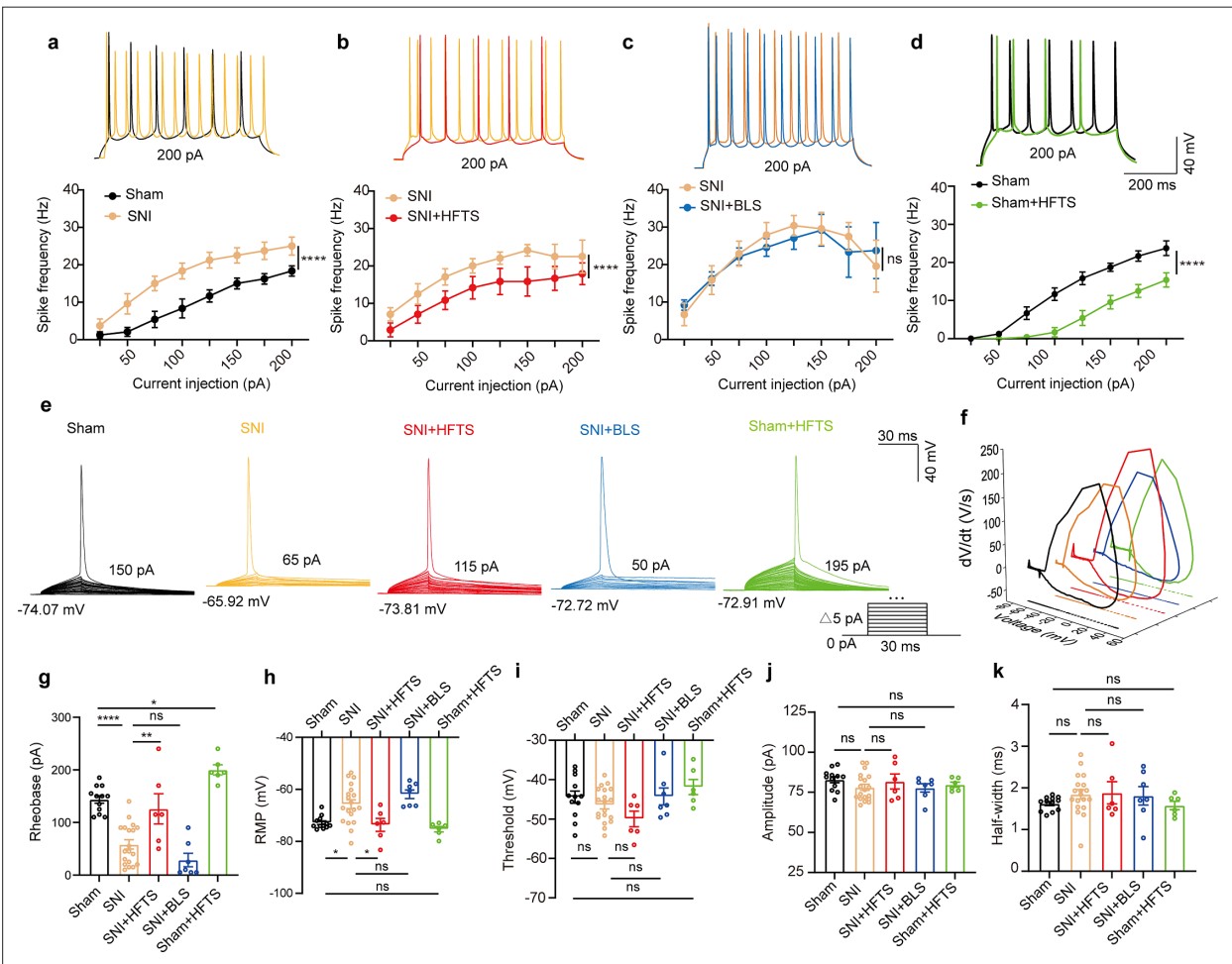

**Figure 3.** High-frequency terahertz stimulation (HFTS) reduces the action potentials (APs) firing rate of pyramidal neurons in spared nerve injury (SNI) and sham mice in vitro. (**a–d**) Representative traces (upper panels) and line charts (lower panels) show the changes of evoked spikes of pyramidal neurons in different groups. (Sham *vs.* SNI: $F_{(1, 40)}$=124.2, p<0.001, $n_{sham}$ = 6, $n_{SNI}$ = 6; SNI *vs.* SNI + HFTS: $F_{(1, 40)}$=23.13, p<0.0001, $n_{SNI}$ = 6, $n_{SNI+HFTS}$=6; SNI *vs.* SNI + BLS: $F_{(1, 40)}$=0.1401, p=0.7101, $n_{SNI}$ = 6, $n_{SNI+BLS}$ = 6; Sham *vs.* Sham + HFTS: $F_{(1, 40)}$=87.29, p<0.0001, $n_{Sham}$ = 6, $n_{Sham+HFTS}$ = 6. Two-way ANOVA followed by *post hoc* comparison using the Šídák's multiple comparisons test). (**e**) Superimposed traces showing the single AP evoked by threshold current stimulation in different groups. (**f**) Phase plots of AP traces in each group. (**g**) Histograms show the statistical comparison of rheobase in each group. (Sham *vs.* SNI: q=8.456, p<0.0001, $n_{sham}$ = 12, $n_{SNI}$ = 19; SNI *vs.* SNI + HFTS: q=5.264, p<0.01, $n_{SNI}$ = 19, $n_{SNI+HFTS}$ = 6; Sham *vs.* Sham + HFTS: q=4.098, p<0.05, $n_{SNI}$ = 19, $n_{SNI+HFTS}$ = 6. one-way ANOVA followed by *post hoc* comparison using the Tukey's multiple comparisons test). (**h**) The resting membrane potential (RMP) in each group (Sham *vs.* SNI: q=4.887, p<0.05, $n_{sham}$ = 12, $n_{SNI}$ = 19; SNI *vs.* SNI +HFTS: q=4.29, p<0.05, $n_{SNI}$ = 19, $n_{SNI+HFTS}$ = 6; Sham *vs.* Sham + HFTS: q=1.261, p>0.05, $n_{SNI}$ = 19, $n_{SNI+HFTS}$ = 6. one-way ANOVA followed by *post hoc* comparison using the Tukey's multiple comparisons test). (**i–k**) HFTS has no significant effect on the threshold, amplitude, and half-width of APs in pyramidal neurons.*p<0.05, **p<0.01, ***p<0.001, ****p<0.0001, ns, p>0.05. BLS, blue light stimulation.

The online version of this article includes the following source data for figure 3:

**Source data 1.** Original data for *Figure 3*.

compared the input-output curves of evoked action potentials before and after HFTS. Our findings revealed a significant increase in the spike frequency in SNI mice (*Figure 3a*; *Supplementary file 2a*), which was effectively rescued by the application of HFTS (*Figure 3b*) (*Supplementary file 2b*), but not by 465 nm blue light stimulation (BLS) (*Figure 3c*; *Supplementary file 2c*). Furthermore, the spike frequency in sham mice also decreased after the application of HFTS (*Figure 3d*) (*Supplementary file 2d*). To further analyze the properties of single action potentials, we induced them by applying a depolarizing current pulse (30 ms) of an appropriate suprathreshold magnitude (*Figure 3e*). In SNI mice, we observed a decrease in the rheobase and an elevation in the resting membrane potential (RMP) compared to those in sham mice. However, these alterations were reversed by the application of HFTS, while BLS had no effects. Other parameters, such as voltage threshold, amplitude, and half-width of the action potentials, were not different between SNI or sham mice with and without HFTS (*Figure 3f–k*). Given that the spike firing, rheobase, and RMP are closely related to low-threshold Kv channels and $K_{Leak}$ channels (*Takeda et al., 2011*; *Tsantoulas and McMahon, 2014*; *González et al., 2012*), these results suggest that HFTS affects the activity of PYR[ACC] neurons through its specific impact on Kv and $K_{Leak}$ channels.

## HFTS decreases the excitability of pyramidal neurons in the ACC in vivo

We then investigate the effect of HFTS on the activities of PYR[ACC] in head-fixed awake SNI mice. One week prior to the illumination experiment, a 16-channel electrode was implanted into the ACC (the detailed structure of this device shows in *Figure 4a*). Then we applied THz photon stimulation for 15 min and compared the neuronal activities before and after HFTS (*Figure 4b*). Our findings revealed a significant decrease in the mean firing rate of ACC neurons after HFTS application in both the sham and SNI groups (*Figure 4c and g*). To further analyze the effect of HFTS on the PYR[ACC], we classified them along with interneurons in the ACC (INT[ACC]) based on their firing rate, trough-to-peak duration, and half-width (*Figure 4d*), as described in our previous study (*Zhu et al., 2022*). We assessed the internal-spiking interval (ISI) and waveform characteristics of the isolated neurons in each channel to ensure that the pre-and post-HFTS units originated from the same neuron (*Figure 4e*). In the sham group, we observed that 63.4% of PYR[ACC] neurons exhibited a decrease in firing rate, 10.8% of PYR[ACC] showed an increase, and 25.8% of PYR[ACC] remained unchanged (93 well-isolated PYR[ACC] neurons were recognized out of 108 total recorded units). In the SNI group, we found that 61.8% of PYR[ACC] neurons exhibited decreased activity, 20.3% of PYR[ACC] showed increased activity, and 17.9% of PYR[ACC] remained unchanged (123 well-isolated PYR neurons were recognized out of 130 total recorded units) (*Figure 4f*). Consistently, the increased mean firing rate of PYR[ACC] neurons in SNI mice was significantly inhibited by the application of HFTS (*Figure 4h*). The activity of INT[ACC] also tended to decrease after HFTS (*Figure 4i*). In contrast, BLS has no effect on the mean firing rate on the PYR[ACC] and INT[ACC] in both sham and SNI mice (*Figure 4—figure supplement 1*). These results indicate that HFTS reduces the spike firing of ACC neurons, whereas BLS does not have the same effect.

## HFTS alleviates mechanical allodynia of SNI mice

Finally, we tested whether applications of HFTS into the ACC induced analgesic effects. The SNI surgery and optic fiber tube implantation into the ACC were performed one week before pain behavioral tests, which included the mechanical pain threshold test and Catwalk analysis (*Figure 5a*). We compared the paw withdrawal mechanical thresholds (PWMTs) before and after HFTS (0.3±0.05 mW at the tip of the optic fiber) application for 15 min and found that SNI treatment significantly decreased the PWMTs compared to the sham group. However, after the application of HFTS, the PWMTs significantly increased, even surpassing those in the sham group in the first 30 min (*Figure 5b*). The analgesic effect lasted for around 160 min with a 15 min application of HTFS and lasted for around 140 min with a 10 min application of HTFS, suggesting a correlation between the duration of analgesia and the intensity of stimulation (*Figure 5c*). In contrast, the PWMTs did not significantly change in the SNI group with the application of 465 nm blue light.

Furthermore, we performed the Catwalk gait analysis (*Figure 5d–l*), which provides exquisite and reliable observations for evaluating the spontaneous pain behaviors (*Zhang et al., 2022b*). We focused on the print intensity and print area-related parameters of the left hind paw (ipsilateral side of the injured nerve). We found that SNI treatment significantly altered the standing time, the stand

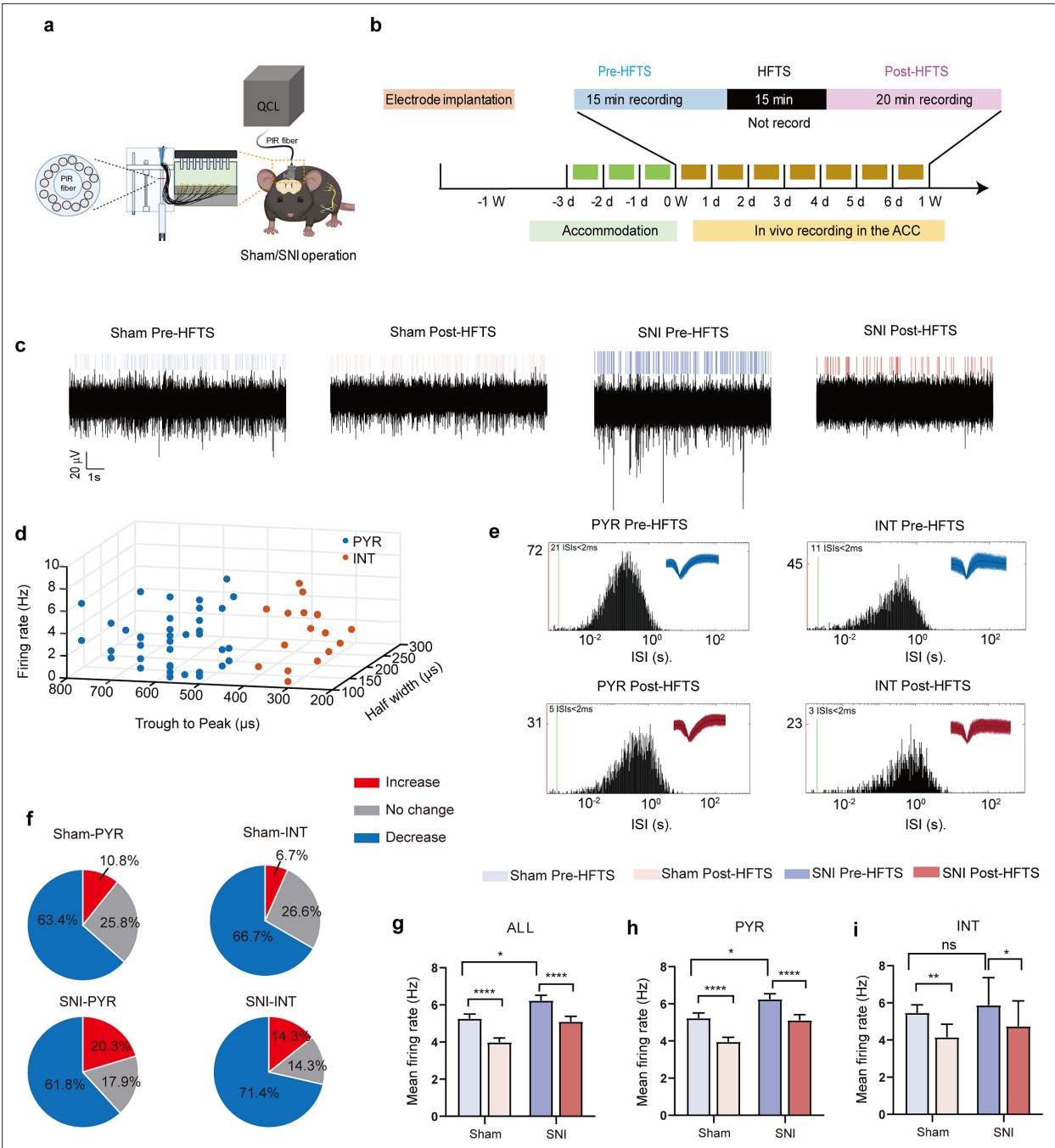

**Figure 4.** High-frequency terahertz stimulation (HFTS) decreases the mean firing rate of pyramidal neurons in the anterior cingulate cortex (ACC) in both sham and spared nerve injury (SNI) awake mice. (**a**) Schematic diagram of the single-unit recording of the ACC using an in vivo multi-channel recording technique. (**b**) The timeline and the stimulating pattern of HFTS on an awake mouse. (**c**) Example recording signals of ACC neurons before and after HFTS application in sham and SNI groups, respectively. (**d**) ACC neurons are classified as pyramidal (PYR) cells and interneurons (INT) using *k*-means cluster-separation algorithm based on their electrophysiological properties. (**e**) Histograms of the inter-spike intervals (ISI) from the spikes of a PYR and an INT in pre-and post-HFTS recording period. Insets at the top right corner show the waveforms of the detected single unit. (**f**) Pie charts summarize the changes in the firing rate of PYR and INT in sham and SNI groups. Pre *vs.* post-HFTS, Wilcoxon rank-sum test. (**g**) The mean firing rate of all recorded neurons in sham and SNI groups before and after HFTS. Sham group (p<0.0001, n=108, Wilcoxon matched-paired signed rank test), SNI group (p<0.0001, n=130, Wilcoxon matched-paired signed rank test), SNI pre-HFTS vs. Sham pre-HFTS (p=0.0447, Mann-Whitney test). (**h**) The mean firing rate of PYR neurons in sham and SNI groups before and after HFTS. Sham group (p<0.0001, n=93, Wilcoxon matched-paired signed rank test), SNI group (p<0.0001, n=123, Wilcoxon matched-paired signed rank test), SNI pre-HFTS vs. Sham pre-HFTS (p=0.0274, Mann-Whitney test). (**i**) The mean firing rate of INT neurons in sham and SNI groups before and after HFTS. Sham group (p=0.0084, n=15, Wilcoxon matched-paired signed rank test), SNI

*Figure 4 continued on next page*

*Figure 4 continued*

group (p=0.0313, n=7, Wilcoxon matched-paired signed rank test), SNI pre-HFTS vs. Sham pre-HFTS (p=0.3322, Mann-Whitney test). *p<0.05, **p<0.01, ****p<0.0001, ns, p>0.05.

The online version of this article includes the following source data and figure supplement(s) for figure 4:

**Source data 1.** Shows the original data from *Figure 4*.

**Figure supplement 1.** Blue light stimulation (BLS) has no effect on the mean firing rate on anterior cingulate cortex (ACC) neurons in both sham and SNI mice.

index, the max contact area, the mean print area, the mean intensity and the duty cycle (*Figure 5g–l*). This suggests that the SNI mice tend to avoid standing and walking on their injured hind paw due to pain hyper sensitivity. The application of HFTS but not BLS rescued most of the above parameters, indicating HFTS' strong analgesic effect.

## Discussion

In the present study, we provide evidence that high-frequency terahertz photons alleviate neuropathic pain in the SNI mice by decreasing the excitability of pyramidal neurons in the ACC. The mechanism underlying this effect is that HFTS increases voltage-gated potassium ion conductance through resonance with the carbonyl group in the potassium channel filter region (*Figure 6*). Unlike optogenetic technology, HFTS can directly regulate the conformation of the ion channel without delivering a transgene that encodes a light-response protein. It exhibits frequency selectivity and dependence on channel structure. This research suggests that HFTS has the potential to serve as a novel optical technology for the treatment of NP pathology.

Neuropathic pain is closely associated with nociceptor excitability in the ACC (*Bliss et al., 2016*; *Zhao et al., 2018*; *Chen et al., 2018a*; *Xu et al., 2008*), with ion channels playing a fundamental role in determining neuronal excitability, particularly in the hyperexcitability of pyramidal neurons (*Trimmer, 2014*). Excitatory Nav channels, responsible for initiating and depolarizing the action potential, can be targeted by inhibitors to effectively decrease or eliminate electrical excitability. These inhibitors are commonly used in neurology as antiepileptic drugs. On the other hand, inhibitory K channels, responsible for repolarization, contribute to the initiation of action potentials in diverse ways (*González et al., 2012*). Enhancing K conductivity could have a similar effect to Nav channel blockers. For instance, retigabine, an activator for Kv7, has recently been approved as a first-in-class antiepileptic drug (*Gao et al., 2022*; *Maljevic and Lerche, 2013*). Among the 12 subfamilies of Kv channels (Kv1-12), Kv1.2 is the most prevalent in neuronal membranes (*Al Sabi et al., 2013*) and has been reported to be significantly associated with neuropathic pain (*Fan et al., 2014*; *Liang et al., 2024*; *Zhang et al., 2021b*). Although the response frequency of Kv1.2 at ~53 THz (*Liu et al., 2021*) or ~34 THz (*Xiao et al., 2023*) and the corresponding modulation function have been verified due to the broad absorption band, this study highlights the significant resonance of Kv1.2 filter structure with photons at 36 THz. By optically stimulating ACC neurons with the frequency of ~36 THz, we observed a significant reduction in the firing rate of pyramidal neurons' action potentials in SNI mice, accompanied by a notable enhancement of $K^+$ conductance. This confirms the effect of THz photons on Kv channels, including Kv1.2. However, to consolidate this conclusion, a more specific pharmacological experiment is necessary. For example, applying a blocking peptide to eliminate the Kv1.2 current and then testing whether this blocks the effects of HTFS would be a valuable test to perform in the future. Moreover, the application of HFTS resulted in significant changes in the rheobase and RMP of pyramidal cells, suggesting that HFTS may also affect the two-pore $K^+$ channels (*Tsantoulas and McMahon, 2014*). We thus tested the effect of HTFS on the $K_{leak}$ current and confirmed this hypothesis. Additionally, it has been reported that basal excitability is influenced by the opening of low-threshold Kv1.2 channels, which filter out small depolarizations and thus control the number of triggered APs (*Al Sabi et al., 2013*). However, we cannot exclude the possibility that THz photons affect other K channel functions, but further research is required to confirm this in the future.

During our research, we focus on studying of pyramidal neurons in the ACC (*Xiao and Zhang, 2018*). It has been reported that the firing rate of glutamatergic pyramidal cells, rather than inhibitory interneurons, increases in the ACC after chronic pain, suggesting an imbalance of excitatory/inhibitory (E/I) ratio (*Zhu et al., 2022*; *Zhao et al., 2018*). In the local circuits of the ACC, inhibitory

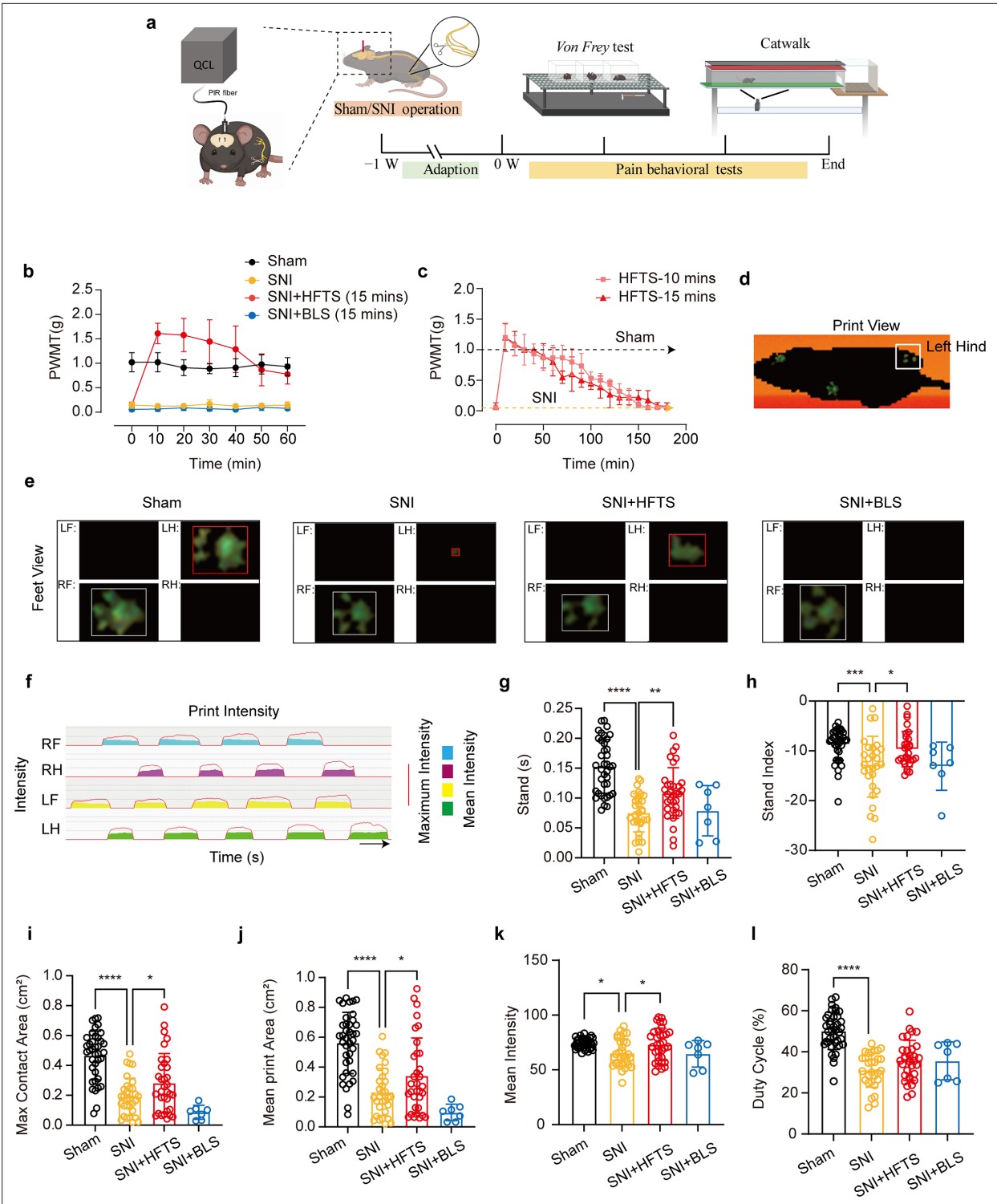

**Figure 5.** High-frequency terahertz stimulation (HFTS) alleviates neuropathic pain of spared nerve injury (SNI) mice through pain behavior tests.
(**a**) Schematic of the establishment of neuropathic pain (NP) model, the application of HFTS in anterior cingulate cortex (ACC) region, and the following behavior tests including *Von Frey* test and Catwalk analysis. (**b**) HFTS increases the paw withdrawal mechanical thresholds (PWMTs) compared to the SNI model ($F_{(18, 140)}$=12.65. p<0.0001. Sham *vs.* SNI: p<0.0001; SNI *vs.* SNI + HFTS: p<0.0001; n=6 in each group. Two-way ANOVA repeated measures followed by *post hoc* comparison using the Šídák's multiple comparisons test). (**c**) Duration of the analgesic effect with HFTS for 10 mins and 15 min. (**d**) The print view of a mouse. (**e**) The feet view of the left front (LF), left hind (LH), right front (RF), and right hind (RH) in the groups of sham, SNI, SNI + HFTS, and SNI + BLS, respectively. (**f**) The step sequence of a sham mouse who passing through the glass pane, the red line represents the maximum

*Figure 5 continued on next page*

*Figure 5 continued*

intensity of each foot, and the color box represents the mean intensity of the corresponding print during walking. (**g**) HFTS increases the LH stand time of SNI mice (sham *vs.* SNI: p<0.0001; SNI *vs.* SNI + HFTS: p<0.01). (**h**) HFTS increases the LH stand index of SNI mice (sham *vs.* SNI: p<0.001; SNI *vs.* SNI + HFTS: p<0.05). (**i**) HFTS increases the LH max contact area of SNI mice (sham *vs.* SNI: p<0.0001; SNI *vs.* SNI + HFTS: p<0.05). (**j**) HFTS increases the LH mean print area of SNI mice (sham *vs.* SNI: p<0.0001; SNI *vs.* SNI + HFTS: p<0.05; SNI vs. SNI + BLS: p<0.05). (**k**) HFTS increases the LH mean intensity of SNI mice (sham *vs.* SNI: p<0.05; SNI *vs.* SNI + HFTS: p<0.05). (**l**) HFTS has no significance for the pain behavior parameter of the duty cycle. *p<0.05, **p<0.01, ***p<0.001, ****p<0.0001. One-way ANOVA (**f–k**) followed by *post hoc* comparison using the Tukey's multiple comparisons test. $n_{Sham}$ = 38, $n_{SNI}$ = 35, $n_{SNI+HFTS}$ = 34, $n_{SNI+BLS}$ = 9.

The online version of this article includes the following source data for figure 5:

**Source data 1.** Provides videos of the behavioral results in mice from each group.

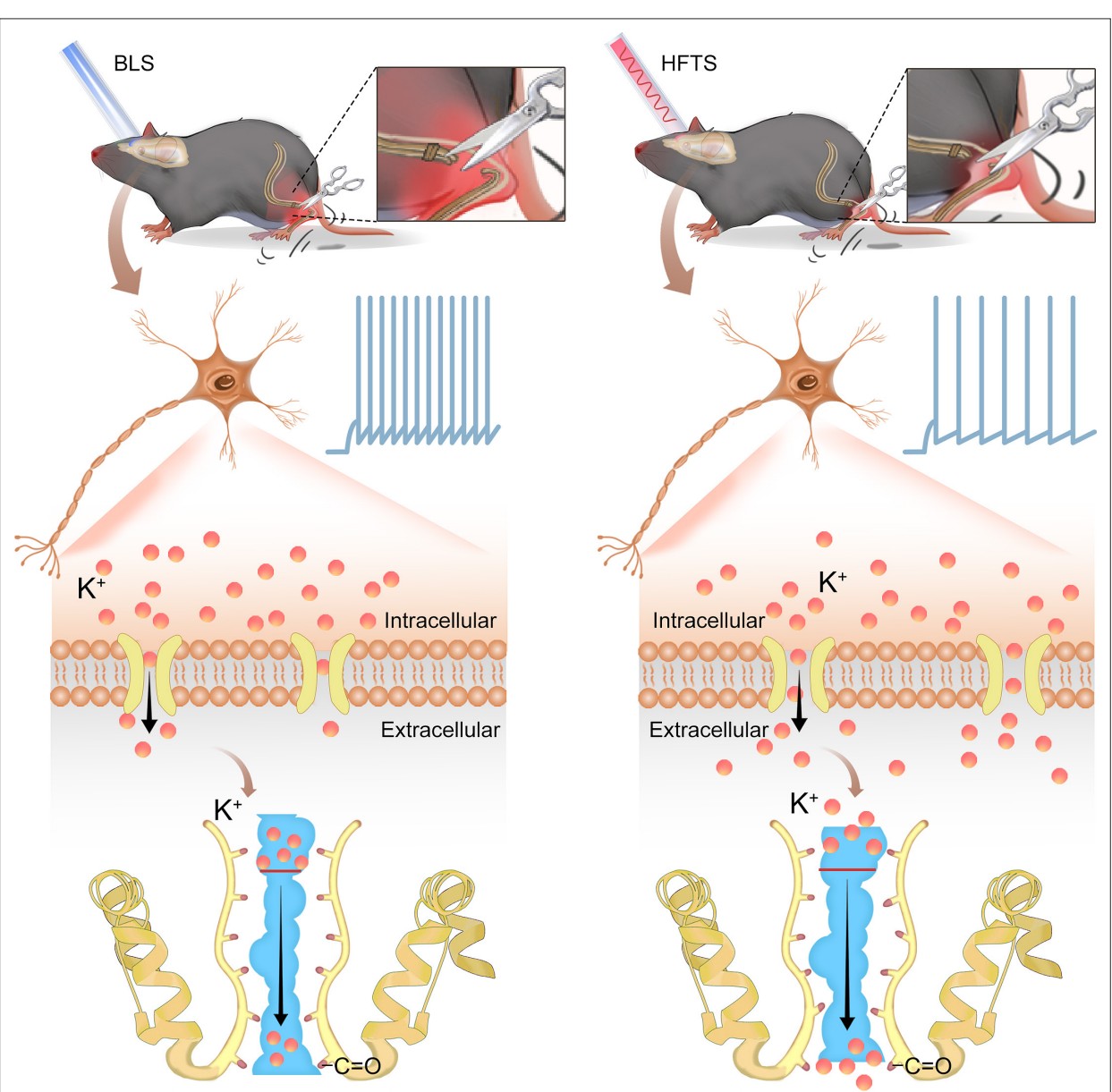

**Figure 6.** Schematic diagram shows the mechanism of high-frequency terahertz stimulation (HFTS) in alleviating neuropathic pain. The left panel shows the group with blue light stimulation (BLS) and the right panel shows the group with HFTS.

neurons release GABA and inhibit the activities of pyramidal cells. Different studies by *Kang et al., 2015* and *Cichon et al., 2017* have reported that specific activation of interneurons in the ACC or in the somatosensory cortex reduces pyramidal neuron hyperactivity and alleviates mechanical allodynia. Thus, the application of THz may induce complicated results by affecting the Kv channels distributed on both pyramidal cells and interneurons. However, as shown in our in vivo recording data (*Figure 4i*), although THz illumination slightly decreased the activity of interneurons, which could potentially lead to an enhanced activity of local pyramidal cells, the direct and significant decrease in pyramidal cell activity caused by the illumination would overcome this disinhibitory effect, ultimately resulting in a net decrease in pyramidal cell activity. The behavioral analgesic effect caused by THz illumination also confirmed this conclusion.

There are several limitations in this study that should be acknowledged. Firstly, we did not investigate the thermal effect of HFTS on the Kv channel. Previous research has demonstrated the non-thermal, long-distance stimulation of high-frequency terahertz stimulation on neuronal activity (*Liu et al., 2021*). Nevertheless, the specific interaction between ~36 THz and Kv channels in terms of thermal effects remains unexplored. Additionally, in this study, we used blue light as a comparison, and found no significant changes in potassium current and the excitability of pyramidal cells. This finding suggests the specificity of the terahertz frequency and supports the existence of non-thermal effects. Another limitation of our research is the use of an optic fiber to deliver the HFTS into the ACC region. This invasive approach may pose challenges for potential noninvasive applications. However, we believe that with the advancement of terahertz enhancement techniques, such as the use of metasurfaces or nanomaterials (*Zhou et al., 2019*; *Wang et al., 2023*), high-frequency terahertz waves show promising potential for broad applications in regulating diverse brain diseases, such as episodic ataxia (*Gao et al., 2022*; *Cole et al., 2021*), benign familial neonatal convulsions (*Jentsch, 2000*), Alzheimer's disease (*Taylor et al., 2022*), and others.

## Materials and methods

### Animals

Male adult (8–10 weeks) C57BL/6 were used for all experiments. Mice were housed on a 12 hr light-dark cycle with food and water freely available. The living conditions were carefully controlled, with temperatures maintained at 22–26°C and humidity at 40%. All animal procedures in the present experiments were in accordance with protocols approved by the Animal Care Committee of the Fourth Military Medical University (IACUC-20210901). All efforts were made to minimize animal suffering and the number of animals used.

### Neuropathic pain model

We used the spared nerve injury (SNI) model to establish neuropathic pain. The detailed process has been described in our previous study (*Zhu et al., 2022*). In brief, mice were generally anaesthetized by 2% isoflurane. Three terminal branches of the left sciatic nerve were exposed by making a direct incision in the skin and a section of the biceps femoris muscle in the left thigh. The tibial nerve and the common peroneal nerve were ligated using 6–0 silk sutures and then sectioned distal to the ligation. After ligating and cutting the nerves, they were carefully put back into their original positions, and the muscles and skin were sutured in two layers. For the sham mice, animals only received an operation that exposed the branches of the left sciatic nerve but without any nerve injury. Following a week's accommodation period, pain behaviors were assessed using the *von Frey* filament test and CatWalk gait analysis to confirm the successful establishment of the NP model.

### Molecular dynamics simulation

The simulation was conducted to gain a deeper understanding of the interaction between terahertz photons and ion channels. A composite model of mouse eukaryotic voltage-gated $K^+$ channels (PDB ID: 3LUT) and eukaryotic $Na^+$ channels (PDB ID: 3RVY) was built using the Charmm-GUI website. The model consisted of intact proteins, phospholipid bilayers, and saline solution (with a concentration of 0.15 M). Kinetic calculations were performed using GROMACS 5.1.2 software. The CHARMM 36 force field and periodic boundary conditions were applied to the proteins. Electrostatic interactions were handled using the connected element algorithm Ewald. During the simulation, the Rattle algorithm

was used to constrain key lengths. The motion equation was solved using the Velocity-Verlet algorithm with a time step of 2 fs. Initially, the simulation was carried out at a room temperature of 303.5 K to observe the ion transport process within the channels at the molecular level. Subsequently, conductivity values (gNa, gK) for potassium and sodium ions and their corresponding absorption spectrum were calculated. To investigate the effect of ion transport under the influence of terahertz radiation, time-varying electric fields of THz radiation were added to the system. In the interaction of terahertz radiation with biological systems, electrical components play a crucial role. The electric field was used to simulate terahertz radiation, and its formula is as follows:

$$E(t) = A \bullet u \bullet \cos(wt + phi)$$

Where A represents the terahertz radiation intensity, u and phi represent the polarization direction and phase of the radiated photon, which are set to (0, 0, 1) and 0, respectively. The terahertz radiation frequency v is related to the angular frequency $\omega$ by the equation:

$$v = \omega/2\pi$$

The cortical neuron Hodgkin-Huxley (H-H) model links the microscopic level of ion channels to the macroscopic level of currents and action potentials. The model consists of two distinct components: a rapid inward current carried by sodium ions and a slower activating outward current carried by potassium ions. These currents result from independent permeability mechanisms for $Na^+$ and $K^+$, where the conductance changes over time and membrane potential. Consequently, the model can replicate and explain a wide range of phenomena, including the shape and propagation of action potentials, the sharp threshold, refractory period, anode-break excitation, accommodation, and subthreshold oscillations. Minor adjustments in key conductance and stimulus current parameters enable the model to describe various action potential phenomena (*Häusser, 2000*). The formula shows as follows:

$$C\frac{dv}{dt} = G_{stim} - g_{Na}(THz)m^3h(v - v_{Na}) - g_K(THz)n(v - v_K) - g_Ln(v - v_L)$$
$$\frac{dy}{dt} = \alpha_y(1 - y) - \beta_y y, \qquad y = m, n$$
$$\frac{dh}{dt} = (\frac{1}{1 + \exp((v + 60)/6.2)} - h)(\alpha_h + \beta_h)$$

Where v, m, h, and n represent the membrane voltage, and probability of the opening or closing of potassium-sodium ion channel. $V_{Na}$, $V_K$, and $V_L$ are the sodium ion reverse potential, potassium ion reverse potential, and resting membrane potential, respectively. $g_{Na}$, $g_K$ are the maximum conductivity of sodium and potassium ions, respectively. C is membrane capacitive reactance with 0.75 uF/cm², the $G_{stim}$ is the stimulation by an external current.

## High-frequency terahertz and blue light stimulation

For HFTS, we used a quantum cascade laser with a center frequency of 35.93±0.1 THz. The laser beam was coupled into a coupler, supported by the Innovation Laboratory of Terahertz Biophysics. We then connected the coupler to a Polycrystalline fiber (PIR) infrared fiber (Art photonics) with a core composition of AgCl/Br. This fiber has excellent transmittance in the range of 3–18 μm, with a core refractive index of 2.15 and an effective numerical aperture (NA) of 0.35±0.05. At the distal end of the fiber, we left approximately 3–4 cm of bare fiber to allow for the insertion of a hollow tube with an inner diameter of 650 μm. This tube was pre-implanted into the ACC region of the SNI and sham group mice brains (*Guo et al., 2014*). The duration of HFTS was 15 min, with a pulse width of 2 μs, a repetition frequency of 10 kHz, and a duty cycle of 40%. The average output power at the tip of the fiber, measured by a MIR detector (NOVA II-3A, Israel), was 0.3±0.05 mW. For comparison purposes, we also used a blue laser to stimulate the same brain region for 15 min, with a frequency of 1 Hz and an average output power of 10 mW.

## In vitro patch clamp recording

The experimental procedures were based on our previous reports (*Zhang et al., 2022a*). Briefly, mice were anesthetized and then decapitated to sacrifice. Brain slices (300 μm thick) containing the ACC were cut on a vibrating microtome (Leica VT 1200 s, Heidelberger, Nussloch, Germany) at 0–4 °C in oxygenated (95% $O_2$ and 5% $CO_2$) artificial cerebrospinal fluid (ACSF) consisting of (in mM) 124 NaCl,

25 NaHCO$_3$, 2.5 KCl, 1 NaH$_2$PO$_4$, 2 CaCl$_2$, 1 MgSO$_4$ and 10 glucose. Slices were then transferred to a room temperature-submerged recovery chamber containing oxygenated ACSF and incubated for at least 1 hr before patch clamp recording. The neurons were then visualized under a microscope with infrared differential interference contrast or fluorescent optics video microscopy (BX51W1, Olympus, Tokyo, Japan). The recording pipettes (3–5 MΩ) were filled with a solution composed of (in mM) 124 K-gluconate, 5 NaCl, 1 MgCl$_2$, 0.2 EGTA, 2 MgATP, 0.1 Na$_3$GTP, 10 HEPES and 10 phosphocreatine disodium (adjusted to pH 7.2 with KOH, 290 mOsmol). Biocytin (0.2%) were added into pippette solution for verifying neurons and visualized through biocytin-avidin reaction. To examine the properties of voltage-gated K$^+$ currents, tetrodotoxin (TTX, 1 µM) and CdCl$_2$ (100 µM) were added into the ACSF. BaCl$_2$ (4 mM) were applied for recording K$_{Leak}$ currents, since barium blocks most K$_{Leak}$ channel subtypes. To examine voltage-gated Na$^+$ current, 3 mM 4-AP and 0.1 mM CdCl$_2$ were added to the ACSF. Electrical signals were filtered at 1 kHz by a Multiclamp 700B amplifier (Molecular Devices, USA), and digitized by an Axon DigiData 1550 A converter with a sampling frequency of 10 kHz. Data analyses were performed with the Clampfit 10.02.

### In vivo multi-channel recording

Before the SNI operation, we implanted an electrode into the right ACC as in our previous reports (*Zhu et al., 2022*), following stereotaxic coordinates: 1.1 mm anterior to the bregma, 0.3 mm lateral to the midline, and 1.8 mm vertical to the skull surface. The electrodes were secured to the exposed skull using the dental adhesive resin cement Super-bond C&B (Japan). This electrode consisted of 16-channel wire electrodes and included a hollow tube. During the optical stimulation, we employed multi-channel recording technology by the Neurolego system (Nanjing Greathink Medical Technology, Nanjing, China). Subsequently, single-unit spike sorting was performed using the MClust-v4.4 toolbox in MATLAB software (MathWorks, USA). In the ACC region, the two main cell types are pyramidal neurons and interneuron cells, which are gamma-aminobutyric acid (GABA) neurons. Pyramidal neurons were primarily classified based on a trough-to-peak duration above 430 µs, indicating long-duration action potentials. Interneuron cells, on the other hand, were identified based on a duration time below 430 µs (*Apkarian et al., 2005*).

### Behavioral assays

#### Mechanical allodynia

Briefly, the paw withdrawal mechanical threshold (PWMT) was evaluated by using von Frey filaments (Stoelting, Kiel, WI, USA) as reported in our previous works (*Zhang et al., 2022a*). Mice were habituated to the testing environment for 3 days before baseline testing and then placed under inverted plastic boxes (7×7×10 cm) on an elevated mesh floor and allowed to habituate for 30 min before threshold testing. A logarithmic series of eight calibrated Semmes-Weinstein monofilaments (von Frey hairs; Stoelting, Kiel, WI, USA) (0.008, 0.02, 0.04, 0.16, 0.4, 0.6, 1, 1.4, and 2 g) with various bending forces (0.078, 0.196, 0.392, 1.568, 3.92, 5.88, 9.8, 13.72, and 19.6 mN) was applied to the plantar surface of the hind paw until the mice withdrew from the stimulus. Positive responses included licking, biting, and sudden withdrawal of the hind paws. A von Frey filament was applied five times (3 s for each stimulus) to each tested area. The minimum bending force of the von Frey filament able to evoke three occurrences of the paw withdrawal reflex was considered the paw withdrawal threshold. All tests were performed in a blinded manner.

#### CatWalk gait analysis

Gait analysis was conducted using the CatWalk XT system (Noldus, the Netherlands) to measure pain-related parameters. The experimental setup involved placing the mouse on a glass platform with open ends, allowing the mouse to walk voluntarily. Simultaneously, a high-speed camera positioned underneath the platform captured images of each step, which were then transmitted to the analysis software (version 10.6, CatWalk XT, Noldus) for further processing. In this study, eight parameters were identified to assess dynamic behaviors relevant to neuropathic pain: Stand: This parameter represents the duration (in seconds) of a paw touching the glass plate; Stand index: it describes the speed at which the paw moves away from the glass plate; Max contact area: it describes the maximum contact area of the paw or leg with the glass plate; Mean print area: it represents the average area of the paw

print during locomotion; Mean intensity: this parameter denotes the average intensity value of the running stage; Duty cycle This parameter denotes the average intensity value of the running stage.

## Statistical analysis

GraphPad Prism 5 (Graph Pad Software, Inc) was used for the statistical analyses and graphing. Statistical significance was assessed by unpaired *t*-test, paired *t*-test, one-way and two-way ANOVA followed by *post hoc* comparison, Wilcoxon matched-paired signed rank test, and Mann-Whitney test. All data in the experiment are expressed in mean ± S.E.M. Statistical significance was indicated as $*p<0.05$, $**p<0.01$, $***p<0.001$ and $****p<0.0001$.

## Acknowledgements

We thank the support of the teaching center of the Air Force medical university for the invaluable technical assistance and J-XG. of our laboratory for the helpful comments on an earlier version of the manuscript and discussions. This work was supported by grants from the National Natural Science Foundation of China (32192410, 32071000 to T C, 82271893 to Y-MW, 8230071226 to W-YP, 82302090 to KC), National Science Fund for Distinguished Young Scholars (12225511 to CC), National Science Fund of China Major Project (T2241002 to CC), Innovation Laboratory of Terahertz Biophysics (23-163-00-GZ-001-001-02-04 to W-YP), The Key Research and Development Plan of Shaanxi Province (S2024-YF-YBSF-0277 to W-YP).

## Additional information

### Funding

| Funder | Grant reference number | Author |
|---|---|---|
| National Natural Science Foundation of China | 32192410 | Tao Chen |
| National Natural Science Foundation of China | 82271893 | Yuanming Wu |
| National Natural Science Foundation of China | 8230071226 | Wenyu Peng |
| National Natural Science Foundation of China | 82302090 | Kun Chen |
| National Science Fund for Distinguished Young Scholars | 12225511 | Chao Chang |
| National Science Fund of China Major Project | T2241002 | Chao Chang |
| Innovation Laboratory of Terahertz Biophysics | 23-163-00-GZ-001-001-02-04 | Wenyu Peng |
| The Key Research and Development Plan of Shaanxi Province | S2024-YF-YBSF-0277 | Wenyu Peng |
| National Natural Science Foundation of China | 32071000 | Tao Chen |

The funders had no role in study design, data collection and interpretation, or the decision to submit the work for publication.

### Author contributions

Wenyu Peng, Data curation, Formal analysis, Funding acquisition, Investigation, Visualization, Writing – original draft, Writing – review and editing; Pan Wang, Data curation, Investigation, Methodology; Chaoyang Tan, Kaijie Wu, Data curation, Investigation, Methodology, Writing – original draft; Han Zhao, Investigation, Methodology; Kun Chen, Funding acquisition, Investigation; Huaxing

Si, Yuchen Tian, Anxin Lou, Methodology; Zhi Zhu, Software, Investigation; Yifang Yuan, Investigation; Chao Chang, Conceptualization, Supervision, Funding acquisition; Yuanming Wu, Supervision, Funding acquisition, Investigation, Methodology; Tao Chen, Conceptualization, Supervision, Funding acquisition, Investigation, Visualization, Methodology, Project administration, Writing – review and editing

## Author ORCIDs
Yuanming Wu ⓘ http://orcid.org/0000-0002-5276-4382
Tao Chen ⓘ https://orcid.org/0000-0003-1956-0553

## Ethics
All animal procedures in the present experiments were in accordance with protocols approved by the Animal Care Committee of the Fourth Military Medical University (Permit Number: IACUC-20210901). All surgery was performed under sodium pentobarbital anesthesia, and every effort was made to minimize suffering.

Reviewer #1 (Public Review): https://doi.org/10.7554/eLife.97444.3.sa1
Reviewer #2 (Public Review): https://doi.org/10.7554/eLife.97444.3.sa2
Reviewer #3 (Public Review): https://doi.org/10.7554/eLife.97444.3.sa3
Author response https://doi.org/10.7554/eLife.97444.3.sa4

## Additional files

### Supplementary files
• Supplementary file 1. Statistical data for *Figure 2*. (a) The Kv current to different voltage in PYR of SNI mice before and after HFTS. (b) Changes of the Kv current impact by HFTS over time. (c) The Kleakcurrent to different voltage in PYR of SNI mice before and after HFTS

• Supplementary file 2. Statistical data for *Figure 3*. (a) The spike frequency in pyramidal (PYR) of anterior cingulate cortex (ACC) before and after spared nerve injury (SNI). (b) The spike frequency in PYR of SNI mice before and after high-frequency terahertz stimulation (HFTS). (c) The spike frequency in PYR of spared nerve injury (SNI) mice before and after blue light stimulation (BLS). (d) The spike frequency in PYR of Sham mice before and after HFTS

• MDAR checklist

• Source code 1. Processing in vivo electrophysiological data code.

### Data availability
All data generated or analysed during this study are included in the manuscript and supporting files; source data files have been provided for Figures 1–5.

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
