## [Editor Report · eLife assessment]

Peng et al. reported **important** findings that 36THz high-frequency terahertz stimulation (HFTS) could suppress the activity of pyramidal neurons by enhancing the conductance of voltage-gated potassium channels. The significance of the findings in this paper is that chronic pain remains a significant medical problem, and there is a need to find non-pharmacological interventions for treatment. The authors present **convincing** evidence that high-frequency stimulation of the anterior cingulate cortex can alter neuronal activity and improve sensory pain behaviors in mice.

---

## [Referee Report · Reviewer #1 (Public Review)]

In this manuscript, by using simulation, in vitro and in vivo electrophysiology, and behavioral tests, Peng et al. nicely showed a new approach for the treatment of neuropathic pain in mice. They found that terahertz (THz) waves increased Kv conductance and decreased the frequency of action potentials in pyramidal neurons in the ACC region. Behaviorally, terahertz (THz) waves alleviated neuropathic pain in the mouse model. Overall, this is an interesting study. The experimental design is clear, the data is presented well, and the paper is well-written.

I have a few suggestions.

(1) The authors provide strong theoretical and experimental evidence for the impact of voltage-gated potassium channels by terahertz wave frequency. However, the modulation of action potential also relies on non-voltage-dependent ion channels. For example, I noticed that the RMP was affected by THz application (Fig. 3F) as well. As the RMP is largely regulated by the leak potassium channels (Tandem-pore potassium channels), I would suggest testing whether terahertz wave photons have also any impact on the Kleak channels as well.

(2) The activation curves of the Kv currents in Fig. 2h seem to be not well-fitted. I would suggest testing a higher voltage (>100 mV) to collect more data to achieve a better fitting.

(3) In the part of behavior tests, the pain threshold increased after THz application and lasted within 60 mins. I suggest conducting prolonged tests to determine the end of the analgesic effect of terahertz waves.

(4) Regarding in vivo electrophysiological recordings, the post-HFTS recordings were acquired from a time window of up to 20 min. It seems that the HFTS effect lasted for minutes, but this was not tested in vitro where they looked at potassium currents. This long-lasting effect of HFTS is interesting. Can the authors discuss it and its possible mechanisms, or test it in slice electrophysiological experiments?

(5) How did the authors arrange the fiber for HFTS delivery and the electrode for in vivo multi-channel recordings? Providing a schematic illustration in Fig. 4 would be useful.

(6) Language is largely OK, but some grammar errors should be corrected.

The authors have completely addressed my concerns. I have no further comments.

---

## [Referee Report · Reviewer #2 (Public Review)]

Summary:

In this manuscript, Peng et al., reported that 36THz high-frequency terahertz stimulation (HFTS) can suppress the activity of pyramidal neurons through enhancing the conductance of voltage-gated potassium channel. The authors also demonstrated the effectiveness of using 36THz HFTS for treating neuropathic pain.

Strengths:

The manuscript is well written and the conclusions are supported by robust results. This study highlighted the potential of using 36THz HFTS for neuromodulation.

Weaknesses:

More characterization of HFTS is needed, so the readers can have a better assessment of the potential usage of HFTS in their own applications.

---

## [Referee Report · Reviewer #3 (Public Review)]

My summary of the manuscript remains the same and is as follows:

This manuscript by Peng et al. presents intriguing data indicating that high-frequency terahertz stimulation (HFTS) of the anterior cingulate cortex (ACC) can alleviate neuropathic pain behaviors in mice. Specifically, the investigators report that terahertz (THz) frequency stimulation widens the selectivity filter of potassium channels thereby increasing potassium conductance leading to a reduction in the excitability of cortical neurons. In voltage clamp recordings from layer 5 ACC pyramidal neurons in acute brain slice, Peng et al. show that HFTS enhances K current while showing minimal effects on Na current. Current clamp recording analyses show that the spared nerve injury model of neuropathic pain decreases the current threshold for action potential (AP) generation and increases evoked AP frequency in layer 5 ACC pyramidal neurons, which is consistent with previous studies. Data are presented showing that ex-vivo treatment with HFTS in slice reduces these SNI-induced changes to excitability in layer 5 ACC pyramidal neurons. The authors also confirm that HFTS reduces excitability of layer 5 ACC pyramidal neurons via in vivo multi-channel recordings from SNI mice. Lastly, the authors show that HFTS is effective at reducing mechanical allodynia in SNI using both the von Frey and Catwalk analyses. Overall, there is considerable enthusiasm for the findings presented in this manuscript given the need for non-pharmacological treatments for pain in the clinical setting.

---

## [Author Response]

The following is the authors’ response to the original reviews.

**Reviewer #1 (Public Review):**
In this manuscript, by using simulation, in vitro and in vivo electrophysiology, and behavioral tests, Peng et al. nicely showed a new approach for the treatment of neuropathic pain in mice. They found that terahertz (THz) waves increased Kv conductance and decreased the frequency of action potentials in pyramidal neurons in the ACC region. Behaviorally, terahertz (THz) waves alleviated neuropathic pain in the mouse model. Overall, this is an interesting study. The experimental design is clear, the data is presented well, and the paper is well-written. I have a few suggestions.(1) The authors provide strong theoretical and experimental evidence for the impact of voltage-gated potassium channels by terahertz wave frequency. However, the modulation of action potential also relies on non-voltage-dependent ion channels. For example, I noticed that the RMP was affected by THz application (Figure 3F) as well. As the RMP is largely regulated by the leak potassium channels (Tandem-pore potassium channels), I would suggest testing whether terahertz wave photons have also any impact on the Kleak channels as well.

Thank you for your positive comment and for providing us with this valuable suggestion. After testing the leak K+ current with and without HFTS on the SNI model, we observed a notable increase in the leak K+ current with HFTS when the holding potential surpassed -40 mV (please see the revised Figs. 2m and n). This finding prompted us to delve deeper into the shifts in the resting membrane potential (RMP). The data, along with statistical analysis, are detailed in Tables S1-3.

(2) The activation curves of the Kv currents in Figure 2h seem to be not well-fitted. I would suggest testing a higher voltage (>100 mV) to collect more data to achieve a better fitting.

Thanks for your advice. We repeated the experiment while maintaining the voltage of patched neurons at a higher level (>100 mV) to collect ample data for better fitting. The outcomes are illustrated in the revised Figs. 2g-j. Clearly, the data reveals a significant increase in K+ conductance in the HFTS group as compared to the SNI group. We have integrated these discoveries into the revised manuscript, replacing the earlier results.

(3) In the part of behavior tests, the pain threshold increased after THz application and lasted within 60 mins. I suggest conducting prolonged tests to determine the end of the analgesic effect of terahertz waves.

Thank you for your insightful comment. We echo your curiosity about the duration of the HFTS effect. In the process of revising our work, we conducted a comparative analysis of the analgesic duration resulting from 10-minute and 15-minute applications of HFTS. The findings are visualized in the revised Fig. 5c. Our observations indicate that after 160 minutes, the PWMT value for the 15-minute HFTS group decreased to a level comparable to that of the SNI group. Meanwhile, the analgesic effects persisted for 140 minutes in the case of the 10-minute HFTS application. These results imply a direct correlation between the duration of HFTS application and the duration of analgesia.

(4) Regarding in vivo electrophysiological recordings, the post-HFTS recordings were acquired from a time window of up to 20 min. It seems that the HFTS effect lasted for minutes, but this was not tested in vitro where they looked at potassium currents. This long-lasting effect of HFTS is interesting. Can the authors discuss it and its possible mechanisms, or test it in slice electrophysiological experiments?

Thank you for your comment. Based on the results from in vivo electrophysiological recordings, it was observed that the effect of HFTS can endure for a minimum of 20 minutes, and this duration was even more extended in behavioral assessments. Taking your advice, we employed slice electrophysiological recording for further testing. Following a 15-minute application of HFTS, we evaluated the K+ current at 5 and 20 minutes after incubation. Our observations clearly indicated a substantial and lasting increase in K+ current, with the effect persisting for at least 20 minutes (refer to Fig. 2l). This provides confirmation of the long-lasting influence of HFTS. The relevant data and statistical analysis are documented in Table S1-2.

(5) How did the authors arrange the fiber for HFTS delivery and the electrode for in vivo multi-channel recordings? Providing a schematic illustration in Figure 4 would be useful.

Thank you for your comment. To enhance the reader's understanding of the HFTS delivery device during multi-channel recording, we have included a schematic illustration in Fig. 4a in the revised manuscript. The top portion of Fig. 4a depicts a quantum cascade laser (QCL) with a center frequency located at approximately 36 THz. This laser is then connected to the recording electrode via a PIR fiber. The left section illustrates the detailed structure of the recording electrode.

(6) Some grammatical errors should be corrected.

Thank you for your thorough review. We have carefully checked and corrected grammar errors we found throughout the entire text to ensure that readers can better comprehend the content of the article.

**Reviewer #2 (Public Review):**
In this manuscript, Peng et al., reported that 36 THz high-frequency terahertz stimulation (HFTS) can suppress the activity of pyramidal neurons by enhancing the conductance of voltage-gated potassium channel. The authors also demonstrated the effectiveness of using 36THz HFTS for treating neuropathic pain.Strengths:The manuscript is well written and the conclusions are supported by robust results. This study highlighted the potential of using 36 THz HFTS for neuromodulation.Weaknesses:More characterization of HFTS is needed, so the readers can have a better assessment of the potential usage of HFTS in their own applications.

Thank you for your suggestion. We have created schematic diagrams illustrating the HFTS delivery (Fig. 4a and Fig. 5a in the revised manuscript). Fig. 4a presents the structure designed for in vivo multi-channel recording. Fig. 5a shows the structure used in behavior test, the recording electrode is replaced by a metal hollow tube, allowing the PIR fiber to pass through the tube and target the ACC region of the mice.

(1) It would be very helpful to estimate the volume of tissue that can be influenced by HFTS. It is not clear how 15 mins HFTS was chosen for this functional study. Does a longer time have a stronger effect? A better characterization of the relationship between the stimulus duration of HFTS and its beneficial effects would be very useful.

Thank you for your feedback. The degree of tissue influence is directly related to the size of the spot emerging from the fiber outlet. In our experiment, we used a PIR fiber with a 630 nm inner core diameter to propagate high-frequency THz waves. This core features a refractive index of 2.15 and has an effective numerical aperture (NA) of 0.35 ± 0.05.

Our decision to apply HFTS for 15 minutes in the behavioral study was primarily based on observations from in vivo multi-channel recordings. Specifically, we noticed a considerable reduction in the average firing rate of PYR cells after 15 minutes of HFTS exposure. To further investigate the correlation between the duration of HFTS stimulation and its effects, we conducted a comparative study using a 10-minute HFTS session. The results, depicted in revised Fig. 5c, reveal that the PWMT value decreased to the level seen in the SNI group after approximately 160 minutes following 15 minutes of HFTS, and after about 140 minutes with 10 minutes of HFTS. This suggests a direct relationship between the length of HFTS application and its beneficial outcomes.

(2) How long does the behavioral effect last after 15 minutes of HFTS? Figure 5b only presents the behavioral effect for one hour, but the pain level is still effectively reduced at this time point. The behavioral measurement should last until pain sensitization drops back to pre-stim level.

Thank you for your feedback. Similar question is also mentioned by reviewer 1. As depicted in Fig. 5c, it was observed that the analgesic effects lasted for 140-160 min with 10-15 minutes application of HFTS. Based on these findings, we can conclude that in the SNI model, targeting the ACC brain region with HFTS for a duration of 10-15 minutes results in an analgesic effect that lasts for roughly 140-160 minutes. This provides valuable insights into the potential clinical applications and duration of relief that can be achieved through HFTS treatment.

(3) Although the manuscript only tested in ACC, it will also be useful to demonstrate the neural modulation effect on other brain regions. Would 36THz HFTS also robustly modulate activities in other brain regions? Or are different frequencies needed for different brain regions?

Thank you for your comment. We hypothesize that light waves at a frequency of approximately 36 THz effectively modulate neuronal activities in various brain regions, primarily due to their impact on K channels. Additionally, we speculate that the application of THz waves at different frequencies may influence other channels, such as Na and Ca channels, potentially facilitating or inhibiting neuronal activities. We believe this is a fascinating and significant area of research to explore in the future.

**Reviewer #3 (Public Review):**
Summary:This manuscript by Peng et al. presents intriguing data indicating that high-frequency terahertz stimulation (HFTS) of the anterior cingulate cortex (ACC) can alleviate neuropathic pain behaviors in mice. Specifically, the investigators report that terahertz (THz) frequency stimulation widens the selectivity filter of potassium channels thereby increasing potassium conductance and leading to a reduction in the excitability of cortical neurons. In voltage clamp recordings from layer 5 ACC pyramidal neurons in acute brain slice, Peng et al. show that HFTS enhances K current while showing minimal effects on Na current. Current clamp recording analyses show that the spared nerve injury model of neuropathic pain decreases the current threshold for action potential (AP) generation and increases evoked AP frequency in layer 5 ACC pyramidal neurons, which is consistent with previous studies. Data are presented showing that ex-vivo treatment with HFTS in slice reduces these SNI-induced changes to excitability in layer 5 ACC pyramidal neurons. The authors also confirm that HFTS reduces the excitability of layer 5 ACC pyramidal neurons via in vivo multi-channel recordings from SNI mice. Lastly, the authors show that HFTS is effective at reducing mechanical allodynia in SNI using both the von Frey and Catwalk analyses. Overall, there is considerable enthusiasm for the findings presented in this manuscript given the need for non-pharmacological treatments for pain in the clinical setting.Strengths:The authors use a multifaceted approach that includes modeling, ex-vivo and in-vivo electrophysiological recordings, and behavioral analyses. Interpretation of the findings is consistent with the data presented. This preclinical work in mice provides new insight into the potential use of directed high-frequency stimulation to the cortex as a primary or adjunctive treatment for chronic pain.Weaknesses:There are a few concerns noted that if addressed, would significantly increase enthusiasm for the study.(1) The left Na current trace for SNI + HFTS in Figure 2B looks to have a significant series resistance error. Time constants (tau) for the rate of activation and inactivation for Na currents would be informative.

Thank you for your feedback. We have carefully considered your comments and made several adjustments in the revised Figs. 2b-f to improve clarity and accuracy. Firstly, we have conducted a comparison of the time constants (tau) between the SNI group and the SNI+HFTS group. These time constants represent the latency of Na current activation or inactivation relative to the half-activated/inactivated voltage. Our analysis reveals that there is no statistically significant difference in tau between the two groups for both activation and deactivation curves. Secondly, we have updated the sample traces in Fig. 2b of the revised manuscript. These new traces illustrate that tau does not significantly differ between the SNI and SNI+HFTS groups, providing a visual representation of our findings. We believe that these modifications strengthen the presentation of our study's details and results, making the data more accessible and understandable for readers.

(2) It is unclear why an unpaired t-test was performed for paired data in Figure 2. Also, statistical methods and values for non-significant data should be presented.

Thank you for your comment. I think you mean the results in Fig. 3. We agree with you that we should use one-way ANOVA to analyze the data since there are more than 2 groups for comparison. We thus re-analyzed the data by using one-way ANOVA in Figs. 3g-k, and have included detailed statistical methods and *P* values in the revised manuscript.

(3) It would seem logical to perform HFTS on ACC-Pyr neurons in acute slices from sham mice (i.e. Figure 3 scenario). These experiments would be informative given the data presented in Figure 4.

Thank you for your valuable advice. During the revision process, we performed HFTS on ACC-PYR neurons in acute slices obtained from sham mice. The findings from this experiment have been integrated into the updated Fig. 3, where the sham group is represented by the green line and histogram (the revised Fig. 3 in the manuscript). It is noteworthy that a significant decrease in spike frequency was observed in the sham mice following HFTS.

(4) As the data are presented in Figure 4g, it does not seem as if SNI significantly increased the mean firing rate for ACC-Pyr neurons, which is observed in the slice. The data were analyzed using a paired t-test within each group (sham and SNI), but there is no indication that statistical comparisons across groups were performed. If the argument is that HFTS can restore normal activity of ACC-Pyr neurons following SNI, this is a bit concerning if no significant increase in ACC-Pyr activity is observed in in-vivo recordings from SNI mice.

Thank you for highlighting the inaccuracies in the analysis. After reviewing the data, we re-analyzed it using alternative statistical methods. In the revised version, since the data did not follow a normal distribution, we employed Wilcoxon matched-paired signed rank tests within the sham and SNI groups, and Mann-Whitney tests between the sham and SNI groups.

Upon comparing the statistical outcomes across the groups, we found that the mean firing rate of 130 ACC neurons in SNI mice was significantly higher compared to that of 108 ACC neurons in sham mice (*P* = 0.0447, Mann-Whitney test). Notably, the mean firing rate of ACC-PYR exhibited a more pronounced increase with a *P* value of 0.0274 in SNI pre-HFTS versus sham pre-HFTS, while the mean firing rate of ACC-INT did not display a significant change across the groups. These findings align with the observations we made in the slice, reinforcing the validity of our results.

(5) The authors indicate that the effects of HFTS are due to changes in Kv1.2. However, they do not directly test this. A blocking peptide or dendrotoxin could be used in voltage clamp recordings to eliminate Kv1.2 current and then test if this eliminates the effects of HFTS. If K current is completely blocked in VC recordings then the authors can claim that currents they are recording are Kv1.1 or 1.2.

Thank you for your kind suggestion. In our research, we employed the Kv1.2 structure as a model to determine the response frequency of terahertz waves. Through both in vitro and in vivo experiments, we were able to demonstrate that the frequency of approximately 36 THz affects the Kv channel and its corresponding spike frequency. Upon analyzing the action potential waveform, we observed a notable variance in the resting membrane potential (RMP). This RMP is predominantly controlled by leak potassium channels, specifically the Tandem-pore potassium channels. In accordance with the recommendation of reviewer 1, we have addressed this particular aspect of our experimentation in the revised manuscript.

We agree that we should use blocking peptides or dendrotoxin to eliminate Kv1.2 current. However, we meet problems in purchasing and delivery of the drugs. We thus added some explanation in the Discussion part to emphasize the value for this pharmacological experiment and can further confirm this in the future works.

(6) The ACC is implicated in modulating the aversive aspect of pain. It would be interesting to know whether HFTS could induce conditioned place preference in SNI mice via negative reinforcement (i.e. alleviation of spontaneous pain due to the injury). This would strengthen the clinical relevance of using HFTS in treating pain.

Thank you for this valuable advice. We share your intrigue regarding this experiment, and we fully recognize the importance and potential of further exploring this area. At present, however, our equipment and platform limitations prevent us from conducting the necessary tests. However, we remain committed to pursuing relevant research opportunities in the future.